# Self-similar chiral organic molecular cages

Zhen Wang [1,2,4] ✉, Qing-Pu Zhang[1,4], Fei Guo[2,4], Hui Ma[1], Zi-Hui Liang[2], Chang-Hai Yi[2], Chun Zhang [1] ✉ & Chuan-Feng Chen [3] ✉

The endeavor to enhance utility of organic molecular cages involves the evolution of them into higher-level chiral superstructures with self-similar, presenting a meaningful yet challenging. In this work, 2D tri-bladed propeller-shaped triphenylbenzene serves as building blocks to synthesize a racemic 3D tri-bladed propeller-shaped helical molecular cage. This cage, in turn, acts as a building block for a pair of higher-level 3D tri-bladed chiral helical molecular cages, featuring multilayer sandwich structures and displaying elegant characteristics with self-similarity in discrete superstructures at different levels. The evolutionary procession of higher-level cages reveals intramolecular self-shielding effects and exclusive chiral narcissistic self-sorting behaviors. Enantiomers higher-level cages can be interconverted by introducing an excess of corresponding chiral cyclohexanediamine. In the solid state, higher-level cages self-assemble into supramolecular architectures of *L*-helical or *D*-helical nanofibers, achieving the scale transformation of chiral characteristics from chiral atoms to microscopic and then to mesoscopic levels.

Organic molecular cages (OMCs)[1–6], a kind of three-dimensional structure with defined cavities and solution-processable properties, have been wildly used in the fields of separation[7–14], catalysis[15–17], and recognition[18]. Due to the widespread utilization of OMCs, numerous OMCs are currently being developed. For now, the construction of OMCs is mainly focused on fabricating a diverse range of OMCs with various geometries, including triangular prisms[8,19–21], cubes[22–25], tetrahedron[26], octahedra[27], and dodecahedra[28,29], etc. by carefully selecting building blocks with different functionalities and shapes. Of interest is that, in recent advancements, scientists have successfully developed intriguing superstructures of interlocked cages by employing non-covalent interactions such as π–π stacking, hydrogen bonding, and coordination bonding among the building blocks[30,31]. Zhang et al. reported another notable cage superstructure featuring twin cavities, named "*diphane*" cages[32,33]. Despite the significant strides made in constructing superstructural OMCs, the current approach primarily focuses on building 3D superstructures with intricate geometric shapes using lower-dimensional organic building blocks. Moreover, there is still a

need to enhance the properties (such as porosity, catalysis, and recognition) of existing OMCs, which currently lag behind polymers like COFs and MOFs[34]. Furthermore, ongoing studies are investigating the construction of supramolecular architectures by employing 3D structural OMCs as monomers to create OMC-based polymers with improved properties[18,35,36]. However, these OMC-based superstructures, in essence, go beyond the scope of organic molecular cages that lose the unique characteristics of solution-processable.

By employing the approach of employing OMCs as foundational units to assemble advanced OMC-based polymers, a strategy is being pursued to develop OMC-based OMC advanced superstructures. Developing a new OMC-based OMC is much more difficult than developing current OMCs as the complexity enhancement in superstructures. Especially, efforts to construct the intricate superstructures displaying self-similarity across different scales in their arrangement for enhancing and expanding their functional properties pose an even greater challenge, requiring meticulous engineering of the constituent building blocks[37,38]. By adopting this strategy, it is anticipated that

[1]College of Life Science and Technology, Huazhong University of Science and Technology, Wuhan 430074, China. [2]National Engineering Laboratory for Advanced Yarn and Fabric Formation and Clean Production, Technology Institute, Wuhan Textile University, Wuhan, Hubei 430200, China. [3]Beijing National Laboratory for Molecular Sciences, CAS Key Laboratory of Molecular Recognition and Function, Institute of Chemistry, Chinese Academy of Sciences, Beijing 100190, China. [4]These authors contributed equally: Zhen Wang, Qing-Pu Zhang, Fei Guo. ✉e-mail: wz@wtu.edu.cn; chunzhang@hust.edu.cn; cchen@iccas.ac.cn

more fascinating structural OMCs could be obtained without compromising the advantageous property of solution processibility.

Chiral self-sorting is a fascinating phenomenon observed in biological systems, where chiral entities selectively interact and organize themselves based on their handedness. Understanding the principles behind chiral self-sorting has far-reaching implications, not only for unraveling the fundamental mechanisms governing life processes but also for designing and engineering advanced materials with tailored chirality and hierarchical organization[39,40]. In recent years, significant progress has been made in the synthesis and design of diverse chiral OMCs through the condensation of imine and boronic ester linkages[41,42]. The good solubility, well-defined structures, ease of characterization, and intrinsic reversibility of bond formation in OMC-based OMCs provide a valuable platform for studying the thermodynamics of cage synthesis and offer opportunities to gain fundamental insights into self-sorting processes[43–45]. Despite the increasing number of studied self-sorting processes during cage formation, where chiral self-sorting OMCs often exhibit a coexistence of both homochiral and heterochiral OMCs[22]. The achievement of exclusive chiral narcissistic self-sorting OMCs, where only one chiral form is selectively formed, remains rare and presents a significant challenge.

Herein, we report a pair of enantiopure higher-level triphenylbenzene (TPB)-based molecular cages 4P-HTMC and 4M-HTMC utilizing TPB-based [2 + 3] O-bridged oxacalixarene molecular cage (TMC)[46,47] as 3D tri-bladed propeller shaped building blocks through dynamic covalent chemistry (DCC). The structural analysis reveals that the 4P-HTMC and 4M-HTMC exhibit elegant 3D tri-bladed helical-shaped structures with hetero-pores (3.9 and 9.8 Å) in one single OMCs, evolving from 3D [2 + 3] OMCs to higher-level [2[2 + 3] + 3] OMCs, which offers an approach to constructing OMCs-based superstructures while preserving their intrinsic cavity and solution processibility. Because of the formation of superstructure[48], these cages showcase an intriguing phenomenon of exclusive narcissistic self-sorting[49,50]. In the solid state, the chiral 4P-HTMC and 4M-HTMC show the supramolecular assembly of chiral L-helical or D-helical nanofibers[51–53], which is challengeable and has not been found yet in OMCs fields. Furthermore, the assembly of 4P-HTMC and 4M-HTMC exhibits the remarkable tunability of polymorphisms, tissue-like assemble structures, *flagellatas*-shaped like vesicles, micro-scaled nanofiber rings[51,54], which give insight evidence for the formation of chiral helical nanofibers. Meanwhile, the emergence of evolved OMCs superstructures may open up possibilities to expand the range of applications.

## Results

### Synthesis of TPB-based [2 + 3] molecular cage (TMC) and structure characterization

The TPB-based [2 + 3] molecular cage (TMC) (Fig. 1 and Supplementary Fig. 1 and 5)[46] was decorated by 4-hydroxybenzaldehyde in N,N-dimethylformamide (DMF) with excess DIPEA at room temperature to afford CHO-TMC. The chemical structure of CHO-TMC was characterized using nuclear magnetic resonance (¹H-NMR and ¹³C-NMR) (Supplementary Figs. 6, 7, and 27) spectroscopy and X-ray single crystal diffraction (SC-XRD).

To obtain suitable well-defined crystals of CHO-TMC for X-ray single crystal diffraction (SC-XRD), CHO-TMC was dissolved in a solution of dioxane and slowly diffused with Et₂O in a closed system over a few days. The single crystals of CHO-TMC were adopted in the monoclinic space group C2/c. With the formation of a molecular cage structure, two TPB scaffolds are linked by three triazine fragments. Due to intermolecular steric hindrance, the free rotation and vibration of the phenyl ring in TPB are partially restricted, resulting in a non-

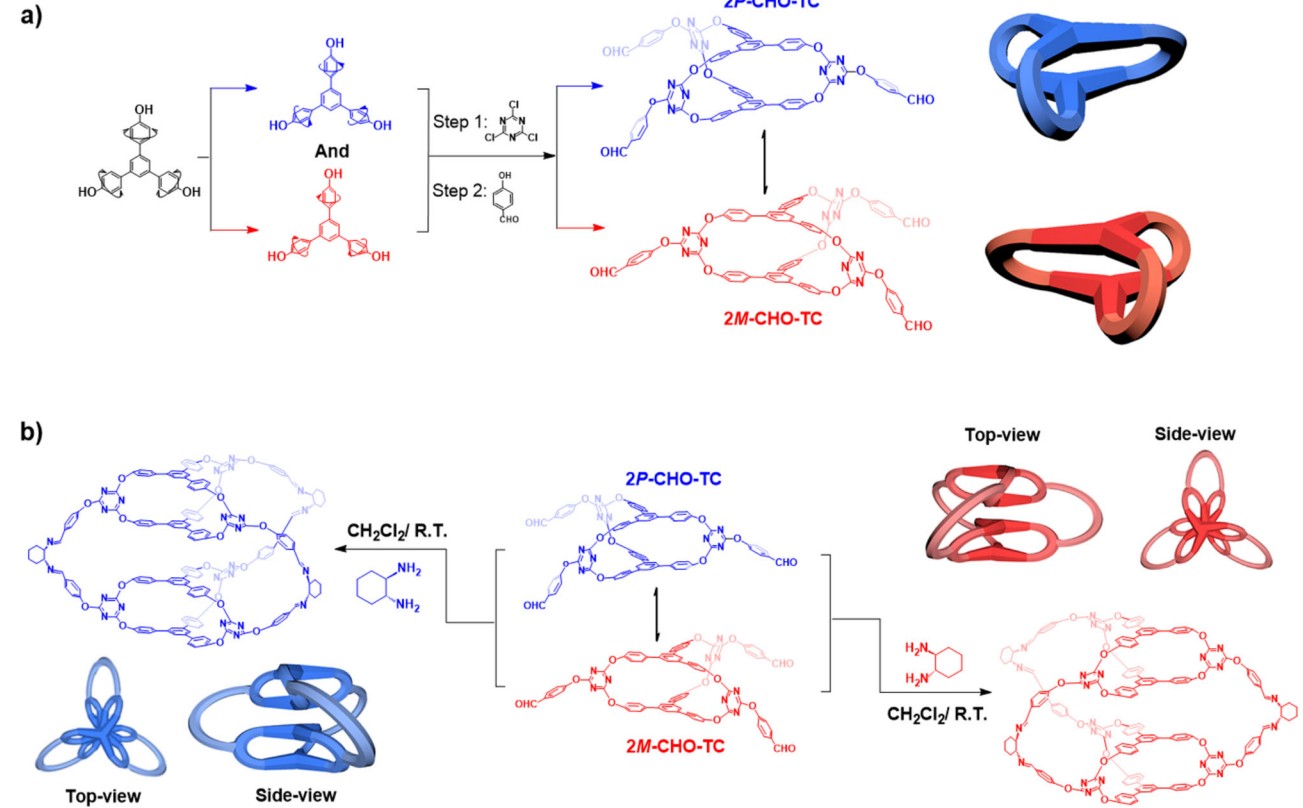

**Fig. 1 | Synthetic route of molecular cages CHO-TMC, 4P-HTMC, and 4M-HTMC. a** Racemic triphenylbenzene (TPB)-based [2 + 3] O-bridged oxacalixarene molecular cages 2P-CHO-TMC (blue) and 2M-CHO-TMC (red). **b** The enantiopure higher-level TPB-based molecular cages 4P-HTMC (blue) and 4M-HTMC (red). For clearly observing the structures, the molecular cages are schematically represented in blue for the (P)-enantiomer, in red for the (M)-enantiomer.

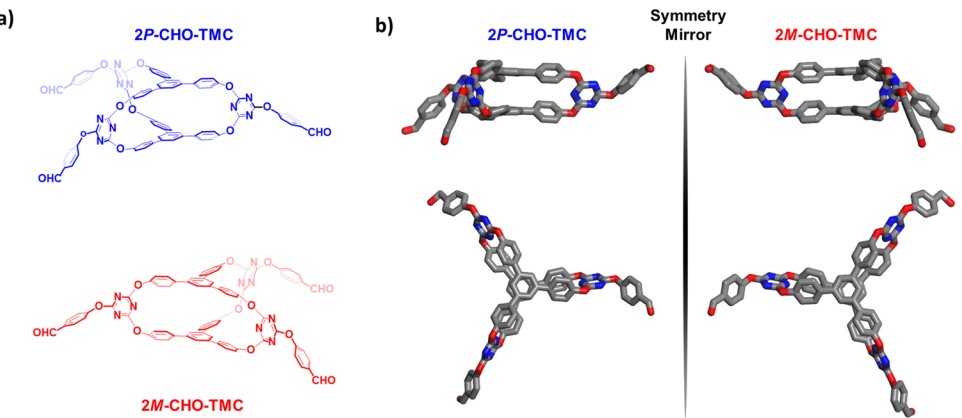

**Fig. 2 | Structures of molecular cage CHO-TMC.** The **a** chemical structure and **b** X-ray single crystal structures of molecular cage CHO-TMC from the side view and top view.

coplanar helical propeller conformation (Supplementary Fig. 30). And, the following opposite rotational directions, two distinct chiral enantiomers, the racemic 2*P*-CHO-TMC and 2*M*-CHO-TMC are formed, exhibiting mirror-symmetric structures (Fig. 2). By computational analysis, the racemic 2*P*-CHO-TMC and 2*M*-CHO-TMCshowed the same total energy of −4395.2800 Ha (Supplementary Fig. 36). In the assembly of the crystal lattice, these enantiomers coexist in equal proportions, resulting in an intertwined network structure (Supplementary Fig. 31). It is essential to highlight that separating these enantiomers proves impractical, primarily due to the partial freedom of rotation and vibration exhibited by the phenyl rings within the molecular cage in solution. This flexibility facilitates their interconversion, a process that lacks any discernible signals in circular dichroism (CD) spectra. (Supplementary Fig. 40). As demonstrated by Zheng's group[55] who successfully achieved the separation of enantiomers with differing rotational directions of the phenyl rings by implementing strategies that restrict and impede the rotation of the phenyl rings. From a structural perspective, the O-bridged [2 + 3] molecular cage TMC (3D) and its scaffold TPB (2D) indeed share a similar tri-bladed propeller-shaped helical structure in different dimensions. Therefore, by incorporating 4-hydroxybenzene into the molecular cage CHO-TMC by the same C−O−C bonds (Supplementary Fig. 32), it becomes possible to utilize this modified cage as a building block for constructing higher-level [2[2 + 3] + 3] molecular cages through a cage-to-cage strategy, where two instances of the modified cage are assembled together to form a larger and more complex molecular cage. The resulting structure would exhibit self-similarity, meaning that it retains the same overall pattern or structure at different scales or levels of magnification.

## Synthesis of higher-level molecular cage 4*P*-HTMC and 4*M*-HTMC and their structure characterizations

For the synthesis of [2[2 + 3] + 3] higher-level molecular cage, the building block racemic CHO-TMC was reacted respectively with (*R*,*R*)- or (*S*,*S*)-diaminocyclohexane (CHDA) in dichloromethane with catalysis amount of trifluoroacetic acid (TFA) at room temperature through DCC (Fig. 1 and Supplementary Figs. 2, 3, and 6–9). Attentionally, regardless of the duration of the reaction or the temperature at which it is conducted, the reaction of molecular cage CHO-TMC remains incomplete until an excess amount of CHDA is added. After the CHO-TMC was consumed by the excess CHDA, the enantiopure higher-level molecular cages 4*P*-HTMC and 4*M*-HTMC were obtained with yields of 94% and 91% by participating with MeOH. The obtained cages 4*P*-HTMC and 4*M*-HTMC were characterized unambiguously by MALDI-TOF mass spectrometry, NMR, and SC-XRD. The results of MALDI-TOF mass spectrometry showed that the 4*P*-HTMC and 4*M*-HTMC calcd for $C_{174}H_{120}N_{24}O_{18}$ [M + H]$^+$: 2835.96, found: 2835.93

(Supplementary Fig. 28 and 29). Compared with the building blocks CHO-TMC, the increased steric hindrance in cages 4*P*-HTMC and 4*M*-HTMC promotes stronger π-π stacking between the aromatic rings, resulting in a higher electron density in the stacked system. As a consequence, the proton signals experience an up-shift in their chemical shifts. Additionally, the proton signals (**d, e, f**) of TPB units in building block CHO-TMC are split into two sets of proton signals (**d, e, f** and **d′, e′, f′**) in cages 4*P*-HTMC and 4*M*-HTMC. Upon analyzing the chemical structures of 4*M*-HTMC and 4*P*-HTMC, they could form intriguing multi-layer sandwich structures, and the proton signals splitting arises from the TPB units in the presence of an inner (**d′, e′, f′**) and outer side of "sandwich" (**d, e, f**). The TPB units located on the inner side exhibit a strong shielding effect and, therefore, lead to a further up-shift of 0.38 ppm (Fig. 3a). To confirm the fascinating phenomenon of intramolecular self-shielding effect[29], the 2D NMR spectra, such as $^1$H,$^1$H-COSY, $^1$H,$^1$H-NOESY, $^1$H,$^{13}$C-HSQC and $^1$H,$^{13}$C-HMBC NMR (Supplementary Figs. 14–21), can provide valuable information about the connectivity and spatial relationships within the molecules, which can help verify the presence of the shielding effect and provide further evidence supporting the observed up-shift and splitting of the proton signals. This unique intramolecular self-shielding effect can hardly be achieved in current OMC systems that only possess single-wall structures, except for some interlocked superstructural OMCs.

The crystals of molecular cages 4*P*-HTMC and 4*M*-HTMC suitable for X-ray single crystal diffraction were obtained by dissolving CHO-TMC molecular cages in a solution of dichloromethane with a catalysis amount of TFA. Then, the solution of CHDA in MeOH was layered on top, and after several days of standing still, needle-shaped crystals appeared at the bottom of the vial. The molecular cages 4*P*-HTMC and 4*M*-HTMC adopted in orthorhombic space group *P* 2$_1$ 2$_1$ 2$_1$, indicating that the structure possesses three mutually perpendicular axes of symmetry and the molecular cages are arranged in a symmetrical manner. Of interest is that the remarkable discovery of higher-level 3D tri-bladed propeller-shaped hierarchical superstructures within the molecular cages 4*P*-HTMC and 4*M*-HTMC unveils a captivating realm of structural intricacies and self-similarity phenomena. What adds a layer of fascination to this observation is the preservation of the analogous 3D tri-bladed propeller-shaped configurations from their molecular cage precursor, CHO-TMC, where the constituent building blocks, TPBs, also exhibited these tri-bladed propeller shapes, as outlined in Fig. 1. This continuity of form across different scales in these OMCs evolutionary process is similar to the evolution of biological macromolecules in nature, where like the protein complexes that form quaternary structures evolved elegant self-similar superstructures, exhibiting self-similarity across different scales in their arrangement that can enhance and expand their functional properties[56].

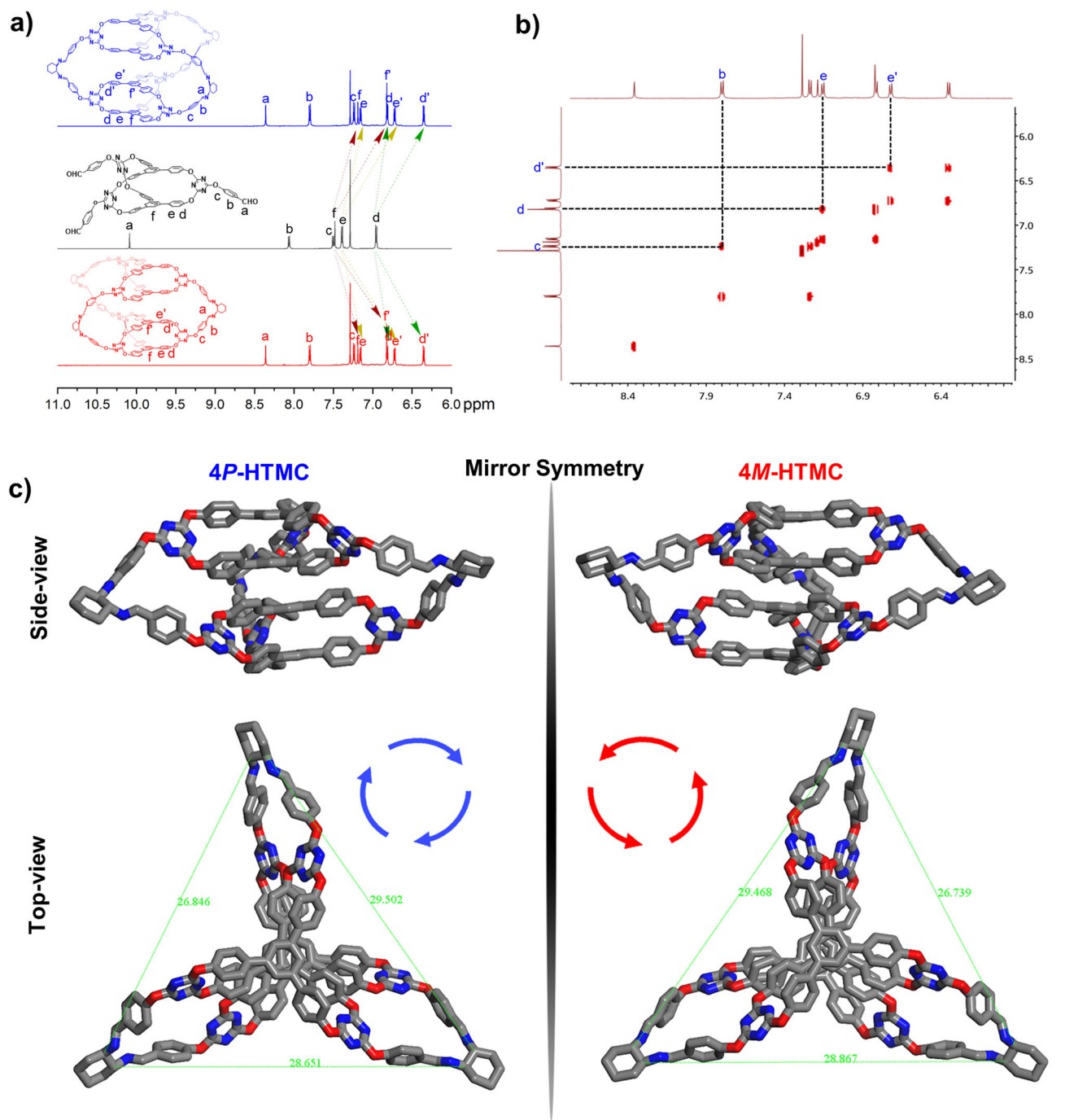

**Fig. 3 | Structural characterization of higher-level molecular cage 4P-HTMC and 4M-HTMC. a** The $^1$H NMR molecular cages of CHO-TMC (black), 4P-HTMC (blue) and 4M-HTMC (red), and **b** the 2D $^1$H,$^1$H-COSY spectrum of 4P-HTMC (600 MHz, CDCl$_3$). **c** X-ray single crystal structures of molecular cages 4P-HTMC and 4M-HTMC from side view and top view. (The hydrogen atom was omitted for clarity).

According to measurements, it has been observed that the distances between the four TPB molecules within the molecular cages 4P-HTMC and 4M-HTMC are 0.2 Å smaller than that in the molecular cage CHO-TMC, which suggests that the four TPB molecules are more compact within molecular cage 4P-HTMC and 4M-HTMC, and implies that there are stronger π-π stacking interactions between them, corresponding to the up-shift in their chemical shifts in NMR (Fig. 3a, S25 and S28). As the higher-level molecular cages 4P-HTMC and 4M-HTMC are formed, they exhibit hetero-pores with a single molecular cage with the defined size of around 3.9 Å and 9.8 Å, which is hardly achieved in current OMCs. The 3.9 Å pore originates from the inherent pore of the building block CHO-TMC, and the pores with around 9.8 Å

are newly introduced (Supplementary Fig. 33c, d). The molecular size of 4P-HTMC and 4M-HTMC is determined to be about 30–34 Å, as measured by the distance between the vertices of the tri-bladed propeller superstructures (Supplementary Fig. 33e, f). Interestingly, these molecular sizes are approximately double and triple larger than their building blocks CHO-TMC and the constituent building blocks TPB, respectively.

The molecular cages 4P-HTMC and 4M-HTMC exhibit a unique and fascinating arrangement, leading to a misalignment of the four TPB molecules in a parallel orientation with different directions due to steric hindrance and the intricate chiral induction and fixation between racemic building blocks CHO-TMC and chiral units CHDA. (Fig. 3c, d,

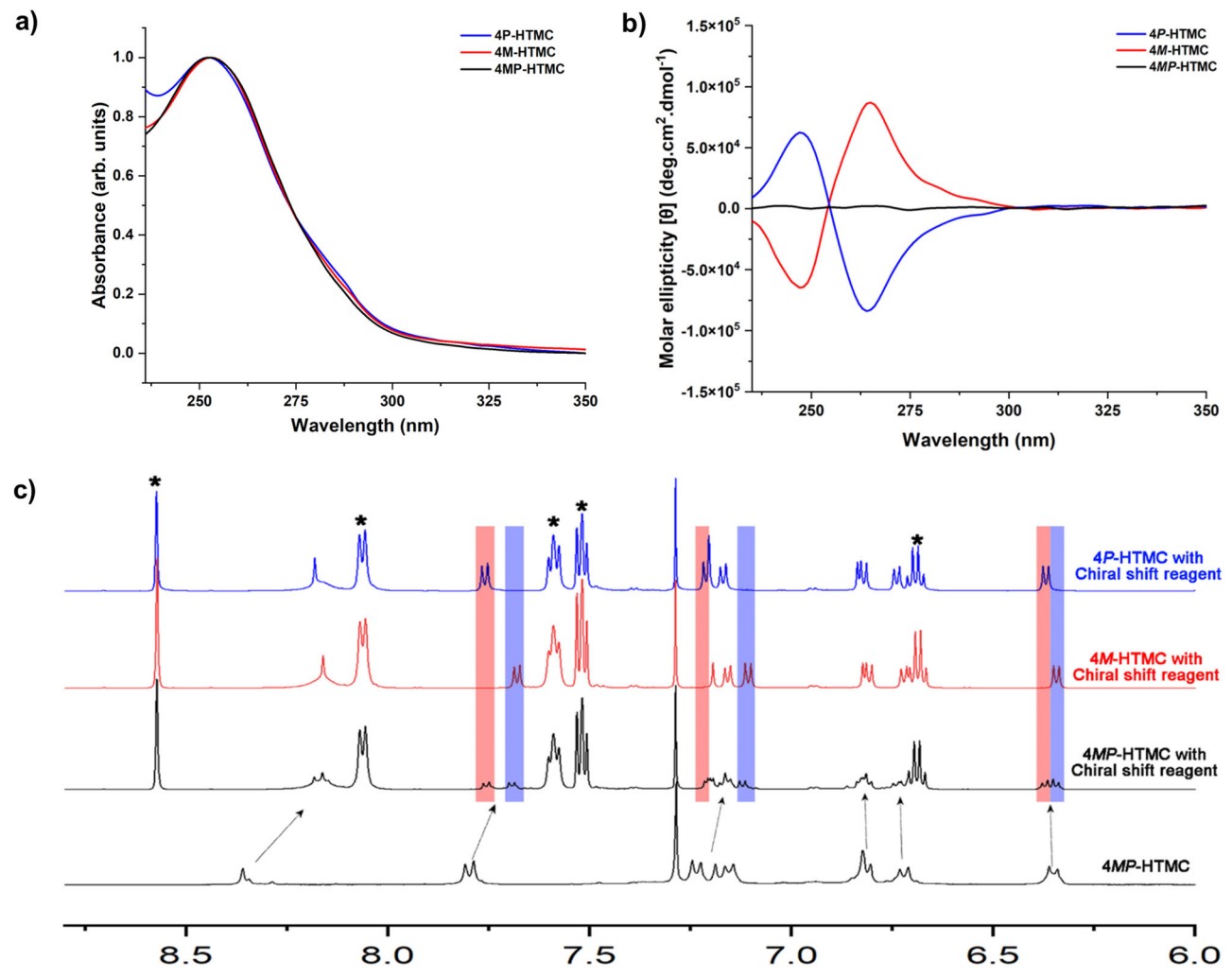

**Fig. 4 | The chiral properties of higher-level molecular cage 4P-HTMC and 4M-HTMC. a** The UV spectra and **b** circular dichroism (CD) spectra of 4*P*-HTMC, 4*M*-HTMC and 4*MP*-HTMC in dichloromethane DCM (*c* = 0.5 mM), and **c** the ¹H NMR (600 MHz, CDCl₃) of molecular cages 4*MP*-HTMC (black), 4*P*-HTMC (blue) and 4*M*-HTMC (red) in CDCl₃ upon adding an excess of the chiral shift reagent (*S*)-( + )−2,2,2-trifluoro-1-(9-anthryl)ethanol.

top view). It is visually demonstrated that molecular cages 4*P*-HTMC and 4*M*-HTMC are a pair of mirror-symmetric tri-helical enantiomers. The restricting rotational direction of the phenyl rings indicates that the chirality of the resulting molecular cages is determined by the chiral (*R,R*)- and (*S,S*)-CHDA. Further, both molecular cages 4*P*-HTMC and 4*M*-HTMC show maximum UV absorption at a wavelength of 250 nm (Fig. 4a) that displayed a 5 nm blue shift as the molecular cage CHO-TMC developed to high-level molecular cages HTMCs (Supplementary Figs. 38 and 39). Accordingly, the circular dichroism (CD) analysis of 4*P*-HTMC and 4*M*-HTMC showed a strong negative Cotton effect in the spectrum of 4*P*-HTMC and a positive mirror-image spectrum for 4*M*-HTMC (Fig. 4b), which can well correspond to their UV spectra. Meanwhile, we also conducted a temperature variable CD experiment. As shown below, through increasing the temperature from 20 to 55 °C, the CD signal intensity of 4*P*-HTMC and 4*M*-HTMC remain unchanged. The results strongly demonstrated that the homochiral HTMC interconversion between *P/M* isomers should be impossible, even at high temperatures (Supplementary Fig. 41).

### Exclusive chiral narcissistic self-sorting behaviors

To study the chiral self-sorting, the molecular cage CHO-TMC was reacted with racemic CHDA to give the product as molecular cage 4*MP*-HTMC (Supplementary Fig. 4). After characterizing with NMR, the 4*MP*-HTMC showed almost the same ¹H-NMR, ¹³C-NMR and 2D NMR

spectra with molecular cages 4*P*-HTMC and 4*M*-HTMC (Supplementary Figs. 12, 13, and 21–25). However, the CD spectrum of the 4MP-HTMC showed no Cotton effect, which means the 4*MP*-HTMC is an achiral compound or racemate. Despite attempting various separation conditions, enantiomeric separation of the molecular cage 4*MP*-HTMC using chiral HPLC was not achieved. By adding an excess of the chiral shift reagent (*S*)-( + )−2,2,2-trifluoro-1-(9-anthryl)ethanol, the ¹H NMR spectrum of 4*MP*-HTMC in CDCl₃ showed pronounced splitting of proton peaks[57]. Under the same conditions with an excess of the chiral shift reagent, by comparing the ¹H NMR spectra of 4*MP*-HTMC with those of 4*P*-HTMC and 4*M*-HTMC, the splitting peaks can exactly match up with the peaks in 4*P*-HTMC and 4*M*-HTMC, respectively (Fig. 4c). For instance, the splitting double peaks at 6.37−6.38 ppm in 4*MP*-HTMC correspond to a specific proton signal found in 4*P*-HTMC, while the splitting double peaks at 6.34−6.35 ppm in 4 *MP*-HTMC correspond to a specific proton signal found in 4*M*-HTMC. Moreover, the relative integration values of these splitting double peaks indicate that the number of protons contributing to each set of peaks is in a 1:1 ratio, which means that there is an equal amount of 4*P*-HTMC and 4*M*-HTMC coexisting in the 4*MP*-HTMC complex. After *Geometry Optimization*, by calculating the four possible isomers HTMC with different ratios of (*R,R*)-(CHDA)/(*S,S*)-(CHDA), which named here with R₃ (4*P*-HTMC), S₃ (4*M*-HTMC) and other two isomers HTMCs (two (*R,R*)-(CHDA) and one (*S,S*)-(CHDA), R₂S) and (one (*R,R*)-(CHDA) and two

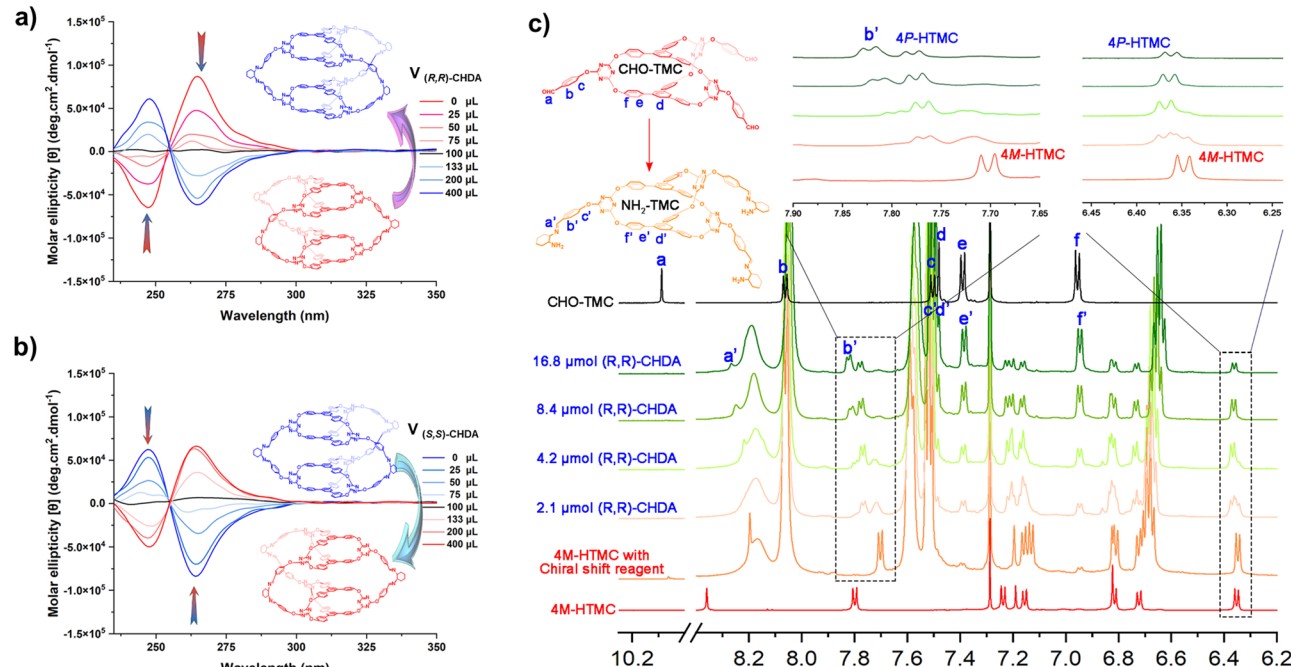

**Fig. 5 | The chiral interconversion properties of higher-level molecular cage 4P-HTMC and 4M-HTMC.** The circular dichroism (CD) spectra of **a** 4M-HTMC and **b** 4P-HTMC in dichloromethane (DCM) (V = 2 mL, c = 0.5 mM) upon adding different volumes of (R,R)-CHDA and (S,S)-CHDA in DCM (c = 10 mM), respectively. **c** The ¹H NMR (600 MHz, CDCl₃) of molecular cages 4M-HTMC with an excess of the chiral shift reagent (S)-( + )−2,2,2-trifluoro-1-(9-anthryl)ethanol in CDCl₃ upon adding different quantity of (R,R)-diaminocyclohexane CHDA.

(S,S)-(CHDA), RS₂) , the total energy of pure chiral HTMC (3R (−9380.4102 Ha) and 3S (−9380.4114 Ha)) are lower than that of hybrid chiral HTMC (R₂S (−9380.3763 Ha, $\Delta E_{(2R+S)\text{-}3S}$ = 22.0 kcal/mol)) and RS₂ (−9380.3805 Ha, $\Delta E_{(2R+S)\text{-}3S}$ = 19.4 kcal/mol)). In pure chiral HTMC (R3 and S3), the misalignment of the four TPB molecules in a parallel orientation can effectively avoid the steric hindrance effects to form a lower energy state with the comparison of hybrid chiral HTMC (R₂S and RS₂) (Supplementary Fig. 37). Therefore, by combining the results above, it can conclude that the 4MP-HTMC is a racemic mixture with an equal amount of 4P-HTMC and 4M-HTMC, which confirm the exclusive chiral narcissistic properties of the HTMC molecular cage (Fig. 4c).

**Chiral interconversion between the 4P-HTMC and 4M-HTMC**
Ascribed to the reversible DCC and the self-sorting property, the chiral conversion between 4P-HTMC and 4M-HTMC was investigated. The successful chiral conversion from 4P-HTMC or 4M-HTMC to their enantiomers can be easily achieved by introducing an enantiomerically opposite chiral CHDA. As shown in Fig. 5a, the results wherein varying amounts of (R,R)-CHDA was added to a solution of 4M-HTMC, along with TFA as a catalyst, and subsequently stirred for several hours. The negative Cotton effects initially observed in the solution of 4M-HTMC gradually diminished and eventually transformed into positive Cotton effects. On the contrary, the positive Cotton effects in the 4P-HTMC solution were transformed into negative Cotton effects with the addition of (S,S)-CHDA (Fig. 5b). These phenomena signified a notable alteration in the chiral structure of the molecular cage. The transition of Cotton effects is a direct indication of the successful chiral conversion that occurred from 4P-HTMC or 4M-HTMC to their enantiomers.

In addition, the ¹H-NMR titration experiments were conducted for further confirmation of the chiral interconversion. As shown in Fig. 5c, focusing on cage 4M-HTMC as a representative case with an excess of the chiral shift reagent, with progressively increased quantity of (R,R)-CHDA, notable changes in the proton signal peaks of 4M-HTMC became evident, while the proton signal peaks of 4P-HTMC exhibited

enhancements. For example, in the spectrum of 4M-HTMC with Chiral shift reagent spectrum, two prominent proton signal peaks in the range of 6.34–6.35 ppm and 7.70–7.71 ppm underwent attenuation with the incremental addition of (R,R)-CHDA. Concurrently, two new proton signal peaks emerged adjacent to these attenuated signals, measuring 6.36–6.37 ppm and 7.77–7.79 ppm, which corresponded to cage 4P-HTMC and displayed intensification. These findings unequivocally demonstrate the remarkable interconversion phenomenon, wherein cage 4M-HTMC transformed into cage 4P-HTMC upon the introduction of an enantiomerically opposite chiral CHDA. Notably, a similar chiral interconversion from 4P-HTMC to cage 4M-HTMC was also observed, as depicted in Supplementary Fig. 26.

However, it is essential to note that as an excess of CHDA into the system, a fraction of the HTMC experienced decomposition, leading to the formation of TMC. Following, the decomposed TMC subsequently reacted with the excess CHDA, giving rise to cages NH₂-TMC, which was supported by titration ¹H-NMR results (Fig. 5c). The addition of CHDA induced the emergence of several new proton signal peaks denoted as a′–e′, which closely corresponded to the proton signal peaks a–e found in the ¹H NMR spectrum of CHO-TMC. The only exception was the proton signal peaks associated with the -CHO group, denoted as peak a, which were evidently involved in the reaction with the excess CHDA, resulting in the formation of NH₂-TMC. As we continued to introduce CHDA, a greater quantity of NH₂-TMC was generated, while more HTMC underwent decomposition, creating a dynamic and intricate interplay within the system. These observations shed light on the complex and fascinating reactions occurring within the HTMC environment, providing further insights into the structural transformations and interconversions facilitated by the presence of chiral CHDA.

**Helical assembly behaviors**
Due to the chiral pairs of 4P-HTMC and 4M-HTMC, they were prone to align in a zig-zag manner to form L- or D-helical structures in the solid state, illustrated by the assembly behavior in their single crystal

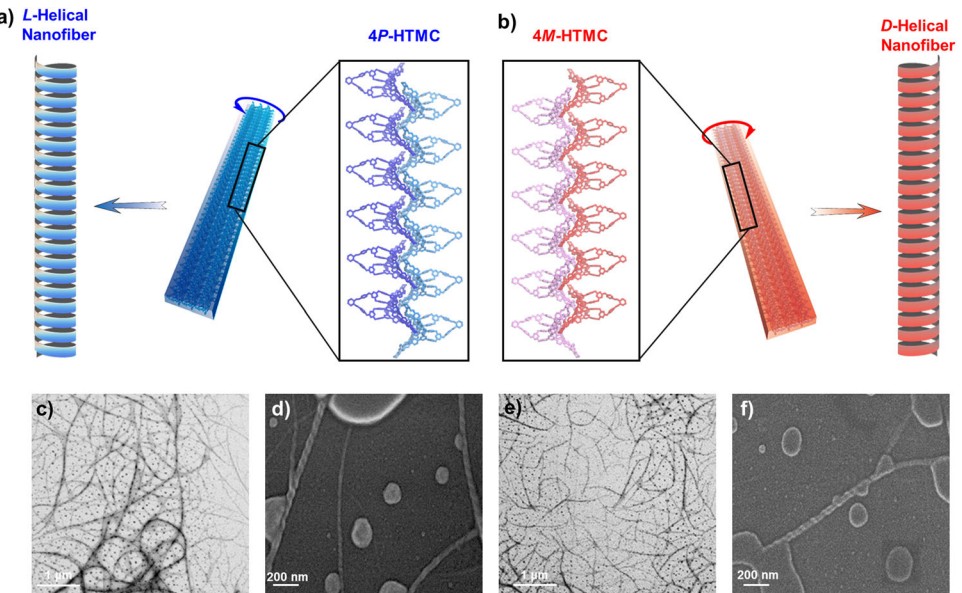

**Fig. 6 | The self-assembly of higher-level molecular cage 4P-HTMC and 4M-HTMC.** The assembly of *L*- or *D*-helical structures by **a** 4*P*-HTMC (blue) and **b** 4*M*-HTMC (red), and the schemes of their assembly process. The transmission electron microscopy (TEM) images of **c** 4*P*-HTMC and **e** 4*M*-HTMC assembled *L*-helical or *D*-helical nanofibers and the scanning electron microscopy (SEM) images of **d** 4*P*-HTMC and **f** 4*M*-HTMC assembled *L*-helical or *D*-helical nanofibers.

assembly (Fig. 6a, b). These structures are composed of repeating units of molecular cages' blades arranged parallel to each other, exhibiting a face-to-face stacking pattern. In this pattern, the intermolecular C−H⋯N interactions occur between the aromatic protons and the nitrogen atoms of the triazine ring, while π−π stacking interactions take place between the triazine ring moieties of adjacent molecular cages (Supplementary Fig. 34). This stacking mode might provide efficient packing and maximize intermolecular interactions, resulting in the stability of the helical structures[58]. Furthermore, through intermolecular interactions, the chiral helical structures assumed by 4*P*-HTMC and 4*M*-HTMC in the solid state are further extended to form 2D layered structures. As the chiral helical structures propagate, neighboring helices align and stack together, resulting in the formation of 2D layered structures. (Supplementary Fig. 35), and then would rotate to form chiral *L*-helical or *D*-helical nanofibers with opposite helical directions.

The assumptions regarding the formation of nanofibers and their chiral properties can be demonstrated through the use of transmission electron microscopy (TEM) and scanning electron microscopy (SEM). To initiate the formation of nanofibers, MeOH is diffused into the solution of 4*P*-HTMC or 4*M*-HTMC in dichloromethane (DCM), resulting in the precipitation of the molecules. The precipitate was dispersed with DCM and ultrasonicated. As shown in Fig. 6c–f, the 4*P*-HTMC or 4*M*-HTMC can assemble into nanofibers with a diameter distribution of 10–50 nm. Careful observation of the surface morphology of the nanofibers shows the presence of *L*- and *D*-helical structures, consistent with their single crystal assembly (Supplementary Fig. 35). The structural insight was further corroborated by powder X-ray diffraction (PXRD) analysis, detailed in Supplementary Fig. 49. The PXRD patterns of 4*P*-HTMC and 4*M*-HTMC exhibited analogous diffraction broad peaks, suggesting a shared assembly mode. The observed broadening of peaks can be attributed to the transition from an ordered assembly to a low crystalline state during the evaporation of solvent molecules from the single crystals. This phenomenon results in a state akin to random assembly between nanofibers, elucidating the similarity in their diffraction patterns. The formation of helical nanofibers can be speculated by the morphology in a successional procession that displayed from tissue-like assembled structures to *flagellatas*-

shaped like vesicles and to micro-scaled nanofiber rings (Supplementary Figs. 44 and 45), which implied that the HTMC assembled membranes curl to helical nanofibers (Supplementary Figs. 46–48). Further, the CD analysis of 4*P*-HTMC and 4*M*-HTMC in the solid state was also investigated. (Supplementary Fig. 42). Compared to 4*P*-HTMC and 4*M*-HTMC in solution, the observed significant red-shift was approximately 30 nm in the CD signals. In solution, the CD signals of 4*P*-HTMC and 4*M*-HTMC primarily arise from the individual chiral cages and their interactions with the surrounding solvent molecules. However, when these molecules assemble in the solid state, the π−π stacking interactions between the aromatic moieties of neighboring molecules become prominent. The enhanced π-π stacking interactions in the solid state lead to a change in the electronic transitions, resulting in the red-shift of the CD signals[59]. This shift in the CD spectrum indicates the formation of chiral aggregates or assemblies where the collective behavior of the molecular cages contributes to the observed chiral response.

In summary, a pair of enantiopure higher-level molecular cages 4*P*-HTMC and 4*M*-HTMC were synthesized by utilizing the racemic TPB-based [2 + 3] O-bridged oxacalixarene molecular cage (TMC) as 3D tri-bladed propeller shaped building blocks through DCC. The X-ray single crystal analysis showed that they exhibit elegant 3D tri-bladed propeller-shaped self-similar superstructures that resembled building blocks CHO-TMC and TPBs across the double and triple-scaled expansion, respectively. The intrinsic cavity (3.9 Å) of building blocks was well preserved, and the new cavity (9.8 Å) was introduced as the cage-based OMCs 4*P*-HTMC and 4*M*-HTMC formed. The unique structural transformation leads to the creation of hetero-pores within a single molecular cage and intramolecular self-shielding effect, a feat that is challenging to achieve in current OMCs. It is indeed intriguing that the assembly of 4*P*-HTMC and 4*M*-HTMC demonstrates exclusive narcissistic self-sorting and chiral inter-transformation, a phenomenon driven by steric hindrance and intricate chiral induction and fixation. In the solid state, they showed a supramolecular assembly of chiral *L*-helical or *D*-helical nanofibers. These findings provide valuable insights into the design and synthesis of OMC-based higher-level supramolecular architectures for improving and expanding applications.

## Methods

### General methods

$^1$H NMR, $^{13}$C NMR, $^1$H,$^1$H-COSY NMR, $^1$H,$^1$H-NOESY NMR, $^1$H,$^{13}$C-HSQC NMR and $^1$H,$^{13}$C-HMBC NMR spectra were recorded on a DMX600 NMR. MALDI-TOF mass spectra were obtained on a BIFLEXIII mass spectrometer. CD spectra were recorded on J-810 Jasco Japan. SEM studies were conducted on JSM6510LV. TEM studies were conducted on a Tecnai G220 electron microscope. The X-ray intensity data were collected on a standard Bruker SMART-1000 CCD Area Detector System equipped with a normal-focus molybdenum-target X-ray tube ($\lambda = 0.71073$ Å) operated at 2.0 kW (50 kV, 40 mA) and a graphite monochromator. The structures were solved by using direct methods and were refined by employing full-matrix least-squares cycles on $F^2$ (Bruker, SHELXTL-97).

### Chiral interconversion between the 4P-HTMC and 4M-HTMC

**4P-HTMC to 4M-HTMC.** CHO-TMC (26 mg, 0.02 mmol) and (R,R)-CHDA (3.3 mg, 0.03 mmol) were dissolved in DCM 20 mL with a catalysis amount of trifluoroacetic acid (TFA). The combined mixture was stirred vigorously at room temperature overnight to afford a 4P-HTMC solution. Then, eight vials were added to the 4P-HTMC solution 2 mL, and different amounts (0, 25, 50, 75, 100, 133, 200, and 400 µL) (S,S)-CHDA (c = 0.3 M) were added in and stirred overnight. The circular dichroism (CD) spectra of them were collected.

**4M-HTMC to 4P-HTMC.** CHO-TMC (26 mg, 0.02 mmol) and (S,S)-CHDA (3.3 mg, 0.03 mmol) were dissolved in DCM 20 mL with a catalysis amount of trifluoroacetic acid (TFA). The combined mixture was stirred vigorously at room temperature for overnight to afford 4M-HTMC solution. Then, eight vials were added to the 4M-HTMC solution 2 mL, and different amounts (0, 25, 50, 75, 100, 133, 200, and 400 µL) (R,R)-CHDA (c = 0.3 M) were added in and stirred overnight. The circular dichroism (CD) spectra of them were collected.

### Helical assembly experiments

**4P-HTMC or 4M-HTMC.** (2 mg) was dissolved in DCM 1 mL in a 5 mL vial and placed in a 100 mL vial with 20 mL MeOH, which resulted in the precipitation of the molecules overnight. The precipitate was dispersed with DCM and ultrasonicated for 30 min. The nanofibers of L- and D-helical structures were formed in the solution. Let stand for 30 min, and the supernatant was dropped on the Cu support films for TEM and SEM.

## Data availability

The authors declare that all other data supporting the findings of this study are available from the article and its Supplementary Information. Source data are provided in this paper. The X-ray crystallographic structures reported in this study have been deposited at the Cambridge Crystallographic Data Centre (CCDC) under deposition numbers 2180996, 2181020, and 2181024. These data can be obtained free of charge from The Cambridge Crystallographic Data Centre via www.ccdc.cam.ac.uk/data_request/cif. Additional data are available from the corresponding author upon request. Source data are provided in this paper.

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

## Acknowledgements

This work is supported by the National Natural Science Foundation of China (22005110, 22031010, and 22275062). We thank Professor Xiang-Gao Meng of Central China Normal University for the SC-XRD test and analysis. We also thank the Analytical and Testing Center of Huazhong University of Science and Technology and the Research Core Facilities for Life Science (HUST) for related analysis.

## Author contributions

Z.W., C.Z., and C.-F.C designed and conceived the study. Z.W., Q.-P.Z., and F.G. did all the experimental work. Z.W. and Q.-P.Z. analyzed and interpreted the results. F.G. was responsible for the NMR experiments. Z.H.L. was responsible for the SEM experiments. H.M. performed *Geometry Optimization*. C.H.Y. contributed essential material and protocols. Z.W. and C.Z. prepared the paper, which was edited by all the authors.

## Competing interests

The authors declare no competing interests.
