## [Peer Review File · Nature Communications]

Reviewers' Comments:

Reviewer #1:

Reviewer's report

- What are the noteworthy results?

The authors present a novel design of an organic molecular cage that facilitates chiral transfer from homochiral CHDA to a racemic TMC subcage, resulting in the formation of a homochiral cage featuring three cavities. This homochirality is attributed to the homochiral CHDA precursor, leading to the discovery of a pair of tri-bladed chiral helical molecular cages that exhibit CD activity. Furthermore, these molecular cages can assemble into chiral helical nanofibers. The authors also demonstrated the ability to tune polymorphisms.

In conclusion, this context showcases a successful instance of chiral transfer within an organic molecular cage and suggests the potential for chiral amplification in hierarchical self-assembly processes.

- Will the work be of significance to the field and related fields? How does it compare to the established literature? If the work is not original, please provide relevant references.

This work holds significant importance within the domains of synthetic chemistry, asymmetric synthesis, and the development of organic porous materials. In previous studies, Zhang *et al.* demonstrated narcissistic self-sorting at the supramolecular level (*J. Am. Chem. Soc.* **2022**, *144*, 1342–1350.), and the same research group showcased the self-sorting of racemic organic cages into conglomerates, which subsequently resolved into homochiral conformers (*Angew. Chem. Int. Ed.* **2023**, *62*, e202217225.). In comparison to these previous works, Chen and his colleagues introduce a novel approach, focusing on chiral transfer from a homochiral precursor to another segment of a complex structure. Furthermore, the study offers insights into remote chiral transfer, demonstrating the transfer of chirality from a point (axial) chirality moiety to planar chirality, adding an intriguing dimension to the research.

- Does the work support the conclusions and claims, or is additional evidence needed?

Current work partially supports the conclusions and claims while quite a number of extra experiments are needed to firmly support the context of this manuscript.

- Are there any flaws in the data analysis, interpretation and conclusions? - Do these prohibit publication or require revision?

No, I don't think there's obvious flaws in the data analysis, interpretation and conclusions.

- Is the methodology sound? Does the work meet the expected standards in your field?

The methodology in the manuscript is generally speaking sound but as mentioned above, it requires additional works to meet the expected standards of a nature communications publication in this field.

- Is there enough detail provided in the methods for the work to be reproduced?

Yes, there're enough details provided in both manuscript and supplementary materials for the work to be reproduced.

- Additional comments

1. In this context, the chirality of HTMC cages is characterized by CD and NMR with the existence of chiral shift reagent. This is not enough to firmly prove homochiral HTMC formation during reaction, authors need to provide chiral-HPLC of as-synthesized sample, enantiomeric excess (ee) value is also needed.
2. In previous work (*Angew. Chem. Int. Ed.* **2023**, *62*, e202217225.), authors found the homochiral cages can interconvert and eventually racemize even at room temperature. In this article, racemic (*P/M*)-CHO-TMC cages are used to synthesize homochiral HTMC cages without bond breaking processes of TMC cages which implicated that the *P* and *M* CHO-TMC cage should be able to interconvert while for homochiral HTMC this interconversion between *P/M* isomers should also be possible, maybe at higher temperature. Hence, a kinetic study to illustrate the kinetic parameters for the racemization processes of both TMC and HTMC cages is important to elaborate the mechanism of chiral transfer process.
3. CD and UV spectra of TMC cages can be helpful to understand later discussion of chiral conversion.
4. In all cases, Cotton effects needed to be assigned.
5. Authors described a chiral conversion from *P* or *M* cage to their enantiomers, however, this process is characterized only by CD. However, while Cotton effects of CD spectra are not assigned, I am not sure whether the two major transition is coming from TMC cage chirality or from only homochiral CHDA imine part. NMR to confirm cages remains as cages after the addition of CHDA is required. An NMR yield characterize the final point of chiral conversion is need. Chiral HPLC traces that proves the transformation of one homochiral species to the other is also important.

Reviewer #2:

Remarks to the Author:

The authors reported the synthesis of innovative organic molecular cages, assembling [2+3] oxacalixarene cages into high-level tri-bladed molecular cage (HTMC). The structures of both [2+3] cages and [2[2+3]+3] HTMC superstructures were well characterized by NMR, MS, CD and XRD. One of the remarkable findings of this study is the "cage to cage" strategy by embedding the intrinsic cavity into the high-level cage entities. In addition, the chiral narcissistic self-sorting behaviors were observed in the presence of racemic CHDA. Furthermore, microscopic HTMC can be further assembled into the macroscopic helical nanofibers, exhibiting a multi-scale chirality transfer phenomenon.

Overall, these findings should be of interest to a broad range of researchers across supramolecular and synthetic chemistry. Upon addressing the following major points, this manuscript is suitable for publication in Nature Communications.

1. The authors should remove the relevant claims regarding the "fractal" cage structures with "self-similarity", which don't align with the mathematical definition and shape of a fractal at all.
2. The authors should carefully revise the introduction to better reflect the study's actual content, refraining from grandiose uncorrelated statements about concepts like "the survival of the fittest in natural selection" and "mimicking evolution". Furthermore, the complexity of organic cages should not be directly compared to the tertiary and quaternary structural complexities in proteins.
3. The author should rephrase claim "these enantiomers coexist in equal proportions..., which lacks any distinguishable signals in circular dichroism (CD) spectra.". The molecular behavior in solution cannot be inferred solely from the solid-state structure. In addition, the indistinguishable CD signal is mainly originated from the unimpeded rotation of phenyl rings in solution on the NMR time scale as indicated by the equivalence of proton (e and f) signals in NMR spectra.
4. The authors should provide the high-resolution MS spectra of CHO-TMC and HTMC.
5. All crystallographic data require further refinement. The authors should address all the CheckCif A and B alerts before re-submitting the CIF file to the CCDC. In addition, the refinement detail should be provided in supplementary information.
6. Considering the intrinsic cavity of CHO-TMC and high-level pore structure of HTMC, the N₂ and CO₂ gas absorption experiments are strongly recommended to showcase their potential for further applications.
7. Taking the propeller conformation of CHO-TMC into account, there are at least 7 possible isomers of HTMC rather than just 4. Analyzing the energy profile of these isomers will be helpful to provide valuable insights into the chiral narcissistic self-sorting behaviors.
8. Does the HTMC decompose into the TMC imine monomers in the presence of excess CHDA during the chiral interconversion? It would be beneficial to conduct the NMR and MS experiments by the progressive addition of CHDA.
9. The Y-axis of CD spectra should be either molar circular dichroism or the molar ellipticity.
10. PXRD experiments of nanofibers are suggested to perform to support the hypothesis that nanofibers have the similar parking structures and assembly behaviors as that in the single crystal structure.

Reviewer #3:

Remarks to the Author:

This is an interesting piece of work presenting the assembly of two tri-bladed cages into a larger tri-bladed structure. The resulting motif within the motif can be considered a fractal although the self-similarity is limited to two levels. TEM studies show that the molecules self-assemble into helical nanofibers. Although these results are interesting there is a plethora of organic cages in the literature that show various shapes and degrees of complexity; the same applies to helical nanofibers.

Key evidence for the structures of reported compounds hinges on X-ray crystallography. It is fair to say that the X-ray work is substandard. R-values are unacceptably high (25%); there is no justification given in the paper or SI as to why they are so high and there are no CheckCIF reports either. A quick check of the supplied CIFs showed long lists of A-alerts. Prior to publication (in any journal) the X-ray work must be brought to an acceptable standard.

REVIEWER COMMENTS

Reviewer #1 (Remarks to the Author):

1. In this context, the chirality of HTMC cages is characterized by CD and NMR with the existence of chiral shift reagent. This is not enough to firmly prove homochiral HTMC formation during reaction, authors need to provide chiral-HPLC of as-synthesized sample, enantiomeric excess (ee) value is also needed.

Thank you for referee's comments. We definitely agree with your suggestion that the chiral-HPLC will help to prove homochiral HTMC formation during reaction. However, we regret to report that, as previously elucidated in our manuscript, our relentless efforts to isolate **4MP-HTMC** have proven exceptionally challenging. Despite our earnest efforts, we conducted an exhaustive exploration of potential solutions within our laboratory. Additionally, we sought external expertise from specialized chiral separation companies, for instance, *DAICEL CHIRAL TECHNOLOGIES (CHINA)CO.LTD* to assist in the separation of the **4M-HTMC** and **4P-HTMC** mixing solution by utilizing a diverse assortment of at least 17 distinct chiral columns. Unfortunately, all endeavors in this regard were met with unyielding difficulties and ultimately did not yield the desired separation outcome.

In our group, our investigation into the molecular cage HTMC remains ongoing, and we are committed to persistently pursuing the isolation of chiral HTMC. If we achieve success in this endeavor, we intend to provide a comprehensive and detailed account of our future work, ensuring that the scientific community is updated on our progress and findings.

2. In previous work (Angew. Chem. Int. Ed. 2023, 62, e202217225.), authors found the homochiral cages can interconvert and eventually racemize even at room temperature. In this article, racemic (P/M)-CHO-TMC cages are used to synthesize homochiral HTMC cages without bond breaking processes of TMC cages which implicated that the P and M CHO-TMC cage should be able to interconvert while for homochiral HTMC

this interconversion between P/M isomers should also be possible, maybe at higher temperature. Hence, a kinetic study to illustrate the kinetic parameters for the racemization processes of both TMC and HTMC cages is important to elaborate the mechanism of chiral transfer process.

Thank you for referee's comments. In response to your valuable suggestion, we have delved into an intriguing recent study (*Angew. Chem. Int. Ed.* 2023, 62, e202217225) and mentioned in our manuscript as reference 41. It focused on the dynamic behavior of homochiral cages, specifically (P)-DC-4 and (M)-DC-4, which display interconversion and racemization phenomena at room temperature. These cages exhibit a distinctive propensity for interconversion, which is hindered by intramolecular hydrogen bonding, ultimately leading each conformer to self-sort into homochiral configurations. It's worth noting that hydrogen bonding, while integral to this behavior, constitutes relatively weak non-covalent interactions that can be easily influenced by external factors such as temperature variations or the presence of guest molecules like lithium diisopropylamide (LDA). This influence of temperature and guest molecules was found to play a pivotal role in driving the interconversion and racemization dynamics observed in these homochiral cages, (P)-DC-4 and (M)-DC-4. Therefore, in this work, the reversible hydrogen bond plays the key factor for the formation of homochiral cages and their chiral interconversion.

Figure 2. Mediation of interconversion rate of (M)- and (P)-DC by intramolecular H-bonding. a) Triple H-bonds of tris(*N*-salicylideneimine) in keto-enamine form can be converted to enolate by deprotonation with lithium diisopropylamide (LDA). b) Slow exchange between (M)- and (P)-DC caused by triple H-bonds (shown with magenta dashed lines) that tether the keto-enamine fragment of the cage, which is dramatically accelerated by attenuation of these H-bonds.

As mentioned in our manuscript, we would like to draw attention to a relevant study

conducted by Zheng et al. In their research, they achieved a noteworthy separation of enantiomers characterized by distinct rotational directions of the phenyl rings within a propeller-like conformation of tetraphenylethylene (TPE). Their approach involved implementing innovative strategies aimed at constraining and hindering the rotational motion of the phenyl rings. Their findings demonstrated that only the homochiral molecule **6**, in which the four phenyl rings in TPE were effectively restricted by the irreversible covalent bonds, exhibited the ability to be successfully separated without undergoing racemization.

Figure 1. Synthetic route of TPE helical molecules **3–6** with stable and metastable propeller-like conformations.

In our work, it is essential to emphasize that the phenyl rings within **CHO-TMC** are not constrained by reversible non-covalent or irreversible covalent bonds, allowing them to freely flip even after the formation of the molecular cage. As a result, the **CHO-TMC** cages possess the ability to freely interconvert between their *P* and *M* conformations. However, when we harnessed the chiral (R,R)-CHDA or (S,S)-CHDA to construct the higher-level molecular cages, **HTMCs**, a crucial transformation occurred. The -NH₂ orientation inherent to the chiral CHDA molecules remained constant, defining the rotation direction of the **HTMC**. This unchangeable orientation, coupled with steric hindrance, played a pivotal role in effectively restricting the rotational motion of the phenyl rings within **CHO-TMC**. This strategic approach was instrumental in preserving the chirality of the resulting **HTMCs**. To illustrate, taking **4M-HTMC** as an example, computational calculations revealed that the total energy of the pure homochiral **4M-HTMC** was -9380.4114 Ha. When we attempted to rotate the

phenyl rings in **4M-HTMC** in the opposite direction while preserving the unalterable C-N bond orientation in chiral CHDA to create **4M-HTMC-R**, a substantial increase in total energy was observed, resulting in -9379.8899 Ha. This energy difference, $\Delta E = 327.2$ kcal/mol, signifies a considerable barrier to the interconversion of phenyl ring rotations in homochiral **HTMC**. Importantly, even when the phenyl rings attempted to rotate in the opposite direction, the orientation of the C-N bond in chiral CHDA remained unaltered.

Meanwhile, we also conducted temperature variable CD experiment. As shown below, through increasing the temperature from 20 to 55 °C, the CD signal intensity of **4P-HTMC** and **4M-HTMC** remain unchanged. The results strongly demonstrated that the homochiral **HTMC** interconversion between *P/M* isomers should be impossible, even at high temperature.

As the results, the **TMC** cage, featuring freely rotating phenyl rings, exhibits an achiral nature in its results. In contrast, the **HTMC** cages, where phenyl rings are fixed or restricted in rotation, display homochirality and remain non-interconvertible even under elevated temperatures. Consequently, we thought that there are no kinetic chiral transfer processes.

Fig. S29 The temperature variable CD spectra of molecular cage **4P-HTMC** (a) and **4M-HTMC** (b) at different 20, 25, 30, 35, 40, 45, 50, 55 °C.

3. CD and UV spectra of TMC cages can be helpful to understand later discussion of chiral conversion.

Thank you for referee's comments. The CD and UV spectra of cage **CHO-TMC** were conducted as shown below, which are added in the supporting information as Figure S32 and S34. These spectra are valuable in enhancing our comprehension of the chiral conversion, as discussed in Question 5 below.

Fig. S32 The UV absorption spectra of molecular cage **CHO-TMC** in DCM ($c = 0.5$ mM).

Fig. S34 The CD spectra of molecular cage **CHO-TMC** in DCM ($c = 0.5$ mM).

4. In all cases, Cotton effects needed to be assigned.

Thank you for referee's comments. Compared with UV spectra of **CHO-TMC**, **4P-HTMC** and **4M-HTMC**, both molecular cages **4P-HTMC** and **4M-HTMC** show maximum UV absorption at wavelength of 250 nm that displayed 5 nm blue shift as the molecular cage **CHO-TMC** developed to high level molecular cages **HTMCs** (Fig S33). The UV absorption of **HTMCs** can well corresponded to their CD spectra. This part is depicted in manuscript as below.

*“Further, both molecular cages **4P-HTMC** and **4M-HTMC** show maximum UV absorption at wavelength of 250 nm (Fig 4a) that displayed 5 nm blue shift as the molecular cage **CHO-TMC** developed to high level molecular cages **HTMCs** (Fig. S33). Accordingly, the circular dichroism (CD) analysis of **4P-HTMC** and **4M-HTMC** showed a strong negative Cotton effect in the spectrum of **4P-HTMC**, and a positive mirror-image spectrum for **4M-HTMC** (Fig. 4b), which can well correspond to their UV spectra.”*

Fig. 4 | The chiral properties of higher-level molecular cage 4P-HTMC and 4M-HTMC. a) the UV spectra and b) circular dichroism (CD) spectra of 4P-HTMC, 4M-HTMC and 4MP-HTMC in DCM ($c = 0.5$ mM).

Fig S33. The UV spectra of 4P-HTMC, 4M-HTMC and 4MP-HTMC in DCM ($c = 0.5$ mM).

5. Authors described a chiral conversion from P or M cage to their enantiomers, however, this process is characterized only by CD. However, while Cotton effects of CD spectra are not assigned, I am not sure whether the two major transition is coming from TMC cage chirality or from only homochiral CHDA imine part. NMR to confirm cages remains as cages after the addition of CHDA is required. An NMR yield characterize the final point of chiral conversion is need. Chiral HPLC traces that proves the transformation of one homochiral species to the other is also important.

Thank you for referee's comments. In the process of chiral interconversion **HTMCs**, the two major transition in CD spectra is definitely coming from the **HTMCs**. As shown in Fig S33 above, by comparing with UV spectra of **CHO-TMC**, **4P-HTMC** and **4M-HTMC**, the maximum UV absorption displayed 5 nm blue shift when the **TMC** developed to **HTMC**. Additionally, as shown the CD spectrum of **CHO-TMC** above (Fig S34), it showed no Cotton effects. Therefore, if the **HTMCs** decompose to **TMC** in the process of chiral interconversion, the Cotton effect would only attenuate but could not convert to opposite Cotton effect, and the Cotton effect would also show blue shift as well. Furthermore, the Cotton effects are unequivocally not from the homochiral CHDA imine, as they lack of UV absorption within the 235-350 nm range. Therefore, it can be concluded that the Cotton effects in CD spectra (Fig. 4c) are originated from the **HTMCs**.

Followed by referee's suggestion, we also conducted NMR titration experiments. Figure 5c illustrates the results of ¹H-NMR titration experiments, focusing on cage **4M-HTMC** as a representative case. In these experiments, we employed an excess of the chiral shift reagent and incrementally introduced (*R,R*)-CHDA. As we progressively increased the quantity of (*R,R*)-CHDA, notable changes in the proton signal peaks of **4M-HTMC** became evident, while the proton signal peaks of **4P-HTMC** exhibited enhancements. To provide a specific example, within the "**4M-HTMC with Chiral shift reagent**" spectrum, two prominent proton signal peaks in the range of 6.34-6.35 ppm and 7.70-7.71 ppm underwent attenuation with the incremental addition of (*R,R*)-CHDA. Concurrently, two novel proton signal peaks emerged adjacent to these attenuated signals, measuring 6.36-6.37 ppm and 7.77-7.79 ppm that corresponded to cage **4P-HTMC** and displayed intensification. These findings unequivocally demonstrate the remarkable interconversion phenomenon, wherein cage **4M-HTMC** transformed into cage **4P-HTMC** upon the introduction of an enantiomerically opposite chiral CHDA. Notably, a similar chiral interconversion from **4P-HTMC** to cage **4M-HTMC** was also observed, as depicted in Fig. S21.

It is essential to note that as an excess of CHDA into the system, a fraction of the **HTMC** experienced decomposition, leading to the formation of **TMC**. Following, the

decomposed **TMC** subsequently reacted with the excess **CHDA**, giving rise to cages **NH₂-TMC**, which was supported by titration ¹H-NMR results. The addition of **CHDA** induced the emergence of several new proton signal peaks denoted as **a'-e'**, which closely corresponded to the proton signal peaks **a-e** found in the ¹H NMR spectrum of **CHO-TMC**. The only exception was the proton signal peaks associated with the -CHO group, denoted as peak **a**, which were evidently involved in the reaction with the excess **CHDA**, resulting in the formation of **NH₂-TMC**. As we continued to introduce **CHDA**, a greater quantity of **NH₂-TMC** was generated, while more **HTMC** underwent decomposition, creating a dynamic and intricate interplay within the system. These observations shed light on the complex and fascinating reactions occurring within the **HTMC** environment, providing further insights into the structural transformations and interconversions facilitated by the presence of chiral **CHDA**.

Fig. 5 | The chiral interconversion properties of higher-level molecular cage 4P-HTMC and 4M-HTMC. The circular dichroism (CD) spectra of a) **4P-HTMC** and b) **4M-HTMC** in DCM ($V=2$ mL, $c = 0.5$ mM) upon adding different volumes of (*S, S*)-CHDA and (*R, R*)-CHDA in DCM ($c = 10$ mM), respectively. c) the ¹H NMR molecular cages of **4M-HTMC** with an excess of the chiral shift reagent (*S*)-(+)-2,2,2-trifluoro-1-(9-anthryl)ethanol in CDCl₃ upon adding different quantity of (*R, R*)-CHDA.

Fig. S21 The ^1H NMR molecular cages of $4P\text{-HTMC}$ with an excess of the chiral shift reagent (S) - $(+)$ -2,2,2-trifluoro-1-(9-anthryl)ethanol in CDCl_3 upon adding different quantity of $(R,R)\text{-CHDA}$.

Reviewer #2 (Remarks to the Author):

The authors reported the synthesis of innovative organic molecular cages, assembling [2+3] oxacalixarene cages into high-level tri-bladed molecular cage (HTMC). The structures of both [2+3] cages and [2[2+3]+3] HTMC superstructures were well characterized by NMR, MS, CD and XRD. One of the remarkable findings of this study is the “cage to cage” strategy by embedding the intrinsic cavity into the high-level cage entities. In addition, the chiral narcissistic self-sorting behaviors were observed in the presence of racemic CHDA. Furthermore, microscopic HTMC can be further assembled into the macroscopic helical nanofibers, exhibiting a multi-scale chirality transfer phenomenon.

Overall, these findings should be of interest to a broad range of researchers across supramolecular and synthetic chemistry. Upon addressing the following major points, this manuscript is suitable for publication in Nature Communications.

1. The authors should remove the relevant claims regarding the "fractal" cage structures with “self-similarity”, which don't align with the mathematical definition and shape of a fractal at all.

Thank you for referee's comments. The term "fractal" was originally coined by the mathematician Benoît Mandelbrot in 1975. He derived it from the Latin word *frāctus*, meaning "broken" or "fractured," and applied it to expand the concept of theoretical fractional dimensions to describe geometric patterns observed in nature. This concept involves the manifestation of similar patterns or structures in nature at progressively smaller scales, a phenomenon known as self-similarity.

In the realm of material science, researchers have utilized the notion of "fractal" to illustrate chemical structures, as demonstrated in a study published in the *Journal of the American Chemical Society* (*J. Am. Chem. Soc.* 2014, 136, 6664–6671, *Hexagon Wreaths: Self-Assembly of Discrete Supramolecular Fractal Architectures Using Multitopic Terpyridine Ligands*). In alignment with this, in our manuscript employed

the term "fractal" to depict the cage-to-cage structure, highlighting self-similarity across different scales.

In response to the referee's suggestion, we have revised our manuscript by removing the mathematical definition of "fractal" and replacing it with the term "self-similar." The high-level molecular cages (**HTMCs**) in our study exhibit a distinctive 3D tri-bladed propeller-shaped structure. Notably, their building blocks, the molecular cage TMCs, also possess a similar 3D tri-bladed propeller structure in a smaller scale. Consequently, we assert that the **HTMCs** can be reasonably characterized as self-similar chiral organic molecular cages.

2. The authors should carefully revise the introduction to better reflect the study's actual content, refraining from grandiose uncorrelated statements about concepts like "the survival of the fittest in natural selection" and "mimicking evolution". Furthermore, the complexity of organic cages should not be directly compared to the tertiary and quaternary structural complexities in proteins.

Thank you for referee's comments. We have diligently revised the introduction in accordance with the provided suggestions. The initial sweeping and uncorrelated statements have been conscientiously eliminated, resulting in a more precise and fitting representation that aligns with the substantive content of the study. The revised introduction now offers a more nuanced and accurate depiction of the research focus, steering clear of grandiose claims. By adhering to the referee's guidance, we have refined the language to ensure a more authentic reflection of the study's scope and objectives. This process has enhanced the overall clarity and coherence of the introduction, providing a solid foundation for readers to engage with the study's content.

3. The author should rephrase claim "these enantiomers coexist in equal proportions..., which lacks any distinguishable signals in circular dichroism (CD) spectra.". The molecular behavior in solution cannot be inferred solely from the solid-state structure. In addition, the indistinguishable CD signal is mainly originated from the unimpeded rotation of phenyl rings in solution on the NMR time scale as indicated by the

equivalence of proton (e and f) signals in NMR spectra.

Thank you for referee's comments. We agree with statements that the molecular behavior in solution cannot be inferred solely from the solid-state structure. So, we rephrase this part in manuscript as below.

“In the assembly of the crystal lattice, these enantiomers coexist in equal proportions, resulting in an intertwined network structure (Fig. S22). It is essential to highlight that separating these enantiomers proves impractical, primarily due to the partial freedom of rotation and vibration exhibited by the phenyl rings within the molecular cage in solution. This flexibility facilitates their interconversion, a process that lacks any discernible signals in circular dichroism (CD) spectra. (Fig. S27).”

4. The authors should provide the high-resolution MS spectra of CHO-TMC and HTMC.

Thank you for referee's comments. The high-resolution MS spectra of CHO-TMC and HTMC have been provided in the Supporting Information

Figure S22. The MALDI-TOF mass spectrometry of CHO-TMC.

Figure S23. The MALDI-TOF mass spectrometry of **4P-HTMC**.

Figure S24. The MALDI-TOF mass spectrometry of **4M-HTMC**.

5. All crystallographic data require further refinement. The authors should address all the CheckCif A and B alerts before re-submitting the CIF file to the CCDC. In addition, the refinement detail should be provided in supplementary information.

Thank you for referee's comments. The **CHO-TMC** underwent a retest in the SC-XRD analysis to enhance the quality of the crystallographic data. Additionally, the crystallographic data for **HTMC** underwent further refinement. All A and B alerts were addressed and resolved, with reasonable explanations provided in the Check-Cif reports. Subsequently, the CIF file was resubmitted to the CCDC.

6. Considering the intrinsic cavity of CHO-TMC and high-level pore structure of HTMC, the N₂ and CO₂ gas absorption experiments are strongly recommended to showcase their potential for further applications.

Thank you for referee's comments. Prior to manuscript submission, gas absorption experiments involving N₂ and CO₂ were conducted, revealing a poor gas adsorption capability characterized by low BET surface areas of 30 or 60 cm²/g. Despite efforts to enhance porous properties, these attempts did not yield significant improvements. Intriguingly, during the dissolution of the solid **HTMC**, a substantial amount of gas bubbles was emitted (As shown below). This phenomenon led us to speculate that **HTMC** has the potential to encapsulate gas molecules through a relatively strong interaction in a tight packing assembly mode. This intriguing observation is the subject of ongoing study, and if feasible, the findings will be reported in our future works.

7. Taking the propeller conformation of CHO-TMC into account, there are at least 7 possible isomers of HTMC rather than just 4. Analyzing the energy profile of these isomers will be helpful to provide valuable insights into the chiral narcissistic self-sorting behaviors.

Thank you for referee's comments. In the formation of **HTMC** with racemic CHDA, there are 4 possible isomers with different ratio of (*R,R*)-(CHDA) and (*S,S*)-(CHDA) without consideration of the propeller conformation of **CHO-TMC**, named (R_3 , R_2S , RS_2 and S_3). After *Geometry Optimization*, the phenyl ring in **CHO-TMC** (including the phenyl rings orientation and angle) in these 4 isomers would adopt a specific conformation for the lowest energy state. In these results, we compared total energy of these 4 isomers, which displayed that the total energy of pure chiral **HTMC** (R_3 and S_3) is lower than hybrid chiral **HTMC** (R_2S and RS_2) by approximately 20 kcal/mol. Taking into account the referee's consideration of the propeller conformation of **CHO-TMC**, for instance, the phenyl rings in **4M-HTMC** were fixed in the opposite direction, resulting in a structure named **4M-HTMC-R**. The total energy of **4M-HTMC-R** is significantly higher than that of **4M-HTMC**, with an energy difference of 327 kcal/mol, indicating the infeasibility of the conformation of **4M-HTMC-R**. Therefore, in the

manuscript, we present energy profiles for the four possible isomers that were optimized to the lowest energy state with their specific propeller conformations.

8. Dose the HTMC decompose into the TMC imine monomers in the presence of excess CHDA during the chiral interconversion? It would be beneficial to conduct the NMR and MS experiments by the progressive addition of CHDA.

Thank you for referee's comments. In the process of chiral interconversion, a fraction of the HTMC experienced decomposition, leading to the formation of TMC. Following, the decomposed TMC subsequently reacted with the excess CHDA, giving rise to cages $\text{NH}_2\text{-TMC}$, which was supported by titration $^1\text{H-NMR}$ results (Fig. 5c). The addition of CHDA induced the emergence of several new proton signal peaks denoted as **a'-e'**, which closely corresponded to the proton signal peaks **a-e** found in the ^1H NMR spectrum of **CHO-TMC**. The only exception was the proton signal peaks associated with the **-CHO** group, denoted as peak **a**, which were evidently involved in the reaction with the excess CHDA, resulting in the formation of **NH₂-TMC**. As we continued to introduce CHDA, a greater quantity of **NH₂-TMC** was generated, while more **HTMC** underwent decomposition, creating a dynamic and intricate interplay within the system.

Fig. 5 | The chiral interconversion properties of higher-level molecular cage 4P-HTMC and 4M-HTMC. The circular dichroism (CD) spectra of a) **4P-HTMC** and b) **4M-HTMC** in DCM ($V=2$ mL, $c = 0.5$ mM) upon adding different volumes of (*S,S*)-CHDA and (*R,R*)-CHDA in DCM ($c = 10$ mM), respectively. c) the ^1H NMR molecular cages of **4M-HTMC** with an excess of the chiral shift reagent (*S*)-(+)-2,2,2-trifluoro-1-(9-anthryl)ethanol in CDCl_3 upon adding different quantity of (*R,R*)-CHDA.

9. The Y-axis of CD spectra should be either molar circular dichroism or the molar ellipticity.

Thank you for referee's comments. The Y-axis of all the CD spectra in the manuscript and the supporting information has been adjusted to represent molar ellipticity. For instance, as shown in Fig. 5a and 5b.

Fig. 5 | The chiral interconversion properties of higher-level molecular cage 4P-HTMC and 4M-HTMC. The circular dichroism (CD) spectra of a) **4P-HTMC** and b) **4M-HTMC** in DCM ($V=2$ mL, $c = 0.5$ mM) upon adding different volumes of (*S, S*)-CHDA and (*R, R*)-CHDA in DCM ($c = 10$ mM), respectively. c) the ^1H NMR molecular cages of **4M-HTMC** with an excess of the chiral shift reagent (*S*)-(+)-2,2,2-trifluoro-1-(9-anthryl)ethanol in CDCl_3 upon adding different quantity of (*R,R*)-CHDA.

10. PXRD experiments of nanofibers are suggested to perform to support the hypothesis that nanofibers have the similar parking structures and assembly behaviors as that in the single crystal structure.

Thank you for referee's comments. The PXRD experiments were performed that the nanofiber and single crystal structure of **HTMCs** displayed similar diffraction peaks as showed below. The broaden diffraction peaks of them were ascribed to the collapse of crystal assembled structure as the solvent evaporated, which is similar to the assembled stacking mode between helical nanofibers. In the manuscript, we have depicted the parts as below.

“Careful observation of the surface morphology of the nanofibers shows the presence of L- and D-helical structures, consistent with their single crystal assembly (Fig. S27). The structural insight was further corroborated by powder X-ray diffraction (PXRD) analysis, detailed in Figure S43. The PXRD patterns of 4P-HTMC and 4M-HTMC exhibited analogous diffraction broad peaks, suggesting a shared assembly mode. The observed broadening of peaks can be attributed to the transition from an ordered assembly to a low crystalline state during the evaporation of solvent molecules from the single crystals. This phenomenon results in a state akin to random assembly between nanofibers, elucidating the similarity in their diffraction patterns.”

Fig. S43 The PXR patterns of a) **4P-HTMC** nanofiber (blue) and crystal solid (black), and b) **4M-HTCM** nanofiber (red) and crystal solid (black).

Reviewer #3 (Remarks to the Author):

This is an interesting piece of work presenting the assembly of two tri-bladed cages into a larger tri-bladed structure. The resulting motif within the motif can be considered a fractal although the self-similarity is limited to two levels. TEM studies show that the molecules self-assemble into helical nanofibers. Although these results are interesting there is a plethora of organic cages in the literature that show various shapes and degrees of complexity; the same applies to helical nanofibers.

Key evidence for the structures of reported compounds hinges on X-ray crystallography. It is fair to say that the X-ray work is substandard. R-values are unacceptably high (25%); there is no justification given in the paper or SI as to why they are so high and there are no CheckCIF reports either. A quick check of the supplied CIFs showed long lists of A-alerts. Prior to publication (in any journal) the X-ray work must be brought to an acceptable standard.

Thank you for referee's comments. The CHO-TMC underwent a retest in the SC-XRD analysis to enhance the quality of the crystallographic data. Additionally, the crystallographic data for HTMC underwent further refinement. All A and B alerts were addressed and resolved, with reasonable explanations provided in the Check-Cif reports. Subsequently, the CIF file was resubmitted to the CCDC.

Reviewers' Comments:

Reviewer #1:

Remarks to the Author:

The authors have invested considerable efforts in responding to reviewer comments by incorporating additional data, explanations, and corrections. It is regrettable that the authors did not successfully employ chiral HPLC to resolve the P/M-HTMC cage, as this could have significantly strengthened the article. Nonetheless, the authors have demonstrated genuine commitment by addressing the raised concerns through alternative methodologies, such as labeled NMRs. In conclusion, while a resolution of the P/M-HTMC cage via chiral HPLC would have enhanced the robustness of the study, the earnest efforts made are evident, and I recommend the publication with only minor corrections.

Additional Comment:

Upon further review, I've noted that the authors have included an energy comparison regarding the chiral transfer within P/M-HTMC. I suggest an additional computational analysis similar to that in Figure 2 for the energy comparison between the enantiomers of CHO-TMC. This would provide readers with valuable insights into how these energy dynamics evolve before and after the formation of the HTMC cage.

Reviewer #3:

Remarks to the Author:

I'm happy with the modifications. Refinements have been improved and checkcif alerts have been addressed.

REVIEWERS' COMMENTS

Reviewer #1 (Remarks to the Author):

The authors have invested considerable efforts in responding to reviewer comments by incorporating additional data, explanations, and corrections. It is regrettable that the authors did not successfully employ chiral HPLC to resolve the P/M-HTMC cage, as this could have significantly strengthened the article. Nonetheless, the authors have demonstrated genuine commitment by addressing the raised concerns through alternative methodologies, such as labeled NMRs. In conclusion, while a resolution of the P/M-HTMC cage via chiral HPLC would have enhanced the robustness of the study, the earnest efforts made are evident, and I recommend the publication with only minor corrections.

Additional Comment:

Upon further review, I've noted that the authors have included an energy comparison regarding the chiral transfer within P/M-HTMC. I suggest an additional computational analysis similar to that in Figure 2 for the energy comparison between the enantiomers of CHO-TMC. This would provide readers with valuable insights into how these energy dynamics evolve before and after the formation of the HTMC cage.

Thank you for referee's comments. Followed your suggestion, we have conducted the computational analysis of the enantiomers *2P*-CHO-TMC and *2M*-CHO-TMC. They showed the same total energy of -4395.2800 Ha, which is almost the half total energy of HTMCs.

Supplementary Fig. 36. The optimized chemical structure and their total energy of four possible molecular cages (a) 2P-CHO-TMC and (b) 2M-CHO-TMC after *Geometry Optimization*. (Hydrogen atom was omitted for clarity).

Reviewer #3 (Remarks to the Author):

I'm happy with the modifications. Refinements have been improved and checkcif alerts have been addressed.

Thank you for referee's comments.